



# Multi-resolution dataset for photovoltaic panel segmentation from satellite and aerial imagery

Hou Jiang[1], Ling Yao[1,2,3,*], Ning Lu[1,2,3], Jun Qin[1,2], Tang Liu[4], Yujun Liu[1,5], Chenghu Zhou[1]

[1]State Key Laboratory of Resources and Environmental Information System, Institute of Geographic Sciences and Natural Resources Research, Chinese Academy of Sciences, Beijing, 100101, China
[2]Southern Marine Science and Engineering Guangdong Laboratory, Guangzhou 511458, China
[3]Jiangsu Center for Collaborative Innovation in Geographical Information Resource Development and Application, Nanjing Normal University, Nanjing 210023, China
[4]School of Information Engineering, China University of Geosciences (Beijing), Beijing, 100083, China
[5]Provincial Geomatics Center of Jiangsu, Nanjing, 210013, China

*Correspondence to*: Ling Yao (yaoling@lreis.ac.cn)

**Abstract.** In the context of global carbon emission reduction, solar photovoltaics (PV) is experiencing rapid development. Accurate localized PV information, including location and size, is the basis for PV regulation and potential assessment of energy sector. Automatic information extraction based on deep learning requires high-quality labelled samples that should be collected at multiple spatial resolutions and under different backgrounds due to the diversity and variable scale of PV. We established a PV dataset using satellite and aerial images with spatial resolutions of 0.8m, 0.3m and 0.1 m, which focus on concentrated PV, distributed ground PV and fine-grained rooftop PV, respectively. The dataset contains 3716 samples of PVs installed on shrub land, grassland, cropland, saline-alkali, and water surface, as well as flat concrete, steel tile, and brick roofs. We used this dataset to examine the model performance of different deep networks on PV segmentation, and on average an intersection over union (IoU) greater than 85% was achieved. In addition, our experiments show that direct cross application between samples with different resolutions is not feasible, and fine-tuning of the pre-trained deep networks using target samples is necessary. The dataset can support more works on PVs for greater value, such as, developing PV detection algorithm, simulating PV conversion efficiency, and estimating regional PV potential. The dataset is available from Zenodo on the following website: https://doi.org/10.5281/zenodo.5171712 (Jiang et al. 2021).

## 1 Introduction

Fossil fuels used by our society have caused unprecedented levels of carbon dioxide ($CO_2$), with widespread climate impacts that threaten human survival and development (Chu and Majumdar 2012; Shin et al. 2021). Therefore, governments around the world intensively made commitments to reduce greenhouse gas emissions and formulated schedules for carbon peak and neutrality. For example, the U.S. government announced the goal of achieving carbon neutrality by 2050, and the Chinese government promised to achieve carbon peak by 2030 and carbon neutrality by 2060. To achieve this, a variety of techniques



have been developed to generate electricity from renewable energy sources (Moutinho and Robaina 2016), of which solar energy has attracted increasing attention because of its endless availability and environmental friendliness (Kabir et al. 2018).

The photovoltaic (PV) market has experienced rapid growth over the past two decades owing to the reduced cost of PV modules and support programs from governments (La Monaca and Ryan 2017; Yan et al. 2019). Between 2000 and 2020,
worldwide installed capacity increased from 4 GW to 714 GW, consistently exceeding expectations (IRENA 2021). Utility-scale PV plants usually need large ground installation area, and face the land use competition with other human activities (Majumdar and Pasqualetti 2019; Sacchelli et al. 2016). Adverse impacts regarding the availability of land resources and erosion are encountered in PV installed regions (Hernandez et al. 2015; Rabaia et al. 2021), which encourages regular monitoring of PV plants during their working lifetime. Distributed solar PVs are installed on marginal agricultural lands
(Martins et al. 2007), building rooftops (Bódis et al. 2019), water surfaces (Liu et al. 2019), and other unused lands, to minimize potential ecological and environmental impacts. In contrast to utility-scale PVs, distributed PVs generate power in isolation; hence, it is necessary to adopt grid-connected technology to integrate them into electrical networks for achieving the greatest benefits (Zambrano-Asanza et al. 2021). To help with PV integration and monitoring, there are strong interests among governments and utility decision-makers in obtaining localized information of existing PVs, such as the location, size,
capacity, and power output (Rico Espinosa et al. 2020; Yao and Hu 2017). Traditional methods, such as in-situ survey and bottom-up reporting, are generally time-consuming and incomplete. In addition, the obtained results lack the desired geospatial precision, and may be outdated due to the rapid growth of PVs. Therefore, frequent data collection is necessary, and efficient data acquisition method is required.

With the advance of spatio-temporal resolution of on-board sensors, satellite and aerial photography can provide
up-to-date images of specific ground targets, making them an ideal source for obtaining accurate PV information (Perez et al. 2001; Peters et al. 2018; Wang et al. 2018). PV panels can be detected and segmented from satellite or aerial images by designing representative features (e.g., color, spectrum, geometry, and texture). However, these features vary with different atmospheric conditions, light circumstances, satellite sensors, observation scales, and surroundings, leading to the defects of generalization ability in extended applications (Ji et al. 2019; Ji et al. 2020; Wang et al. 2018). Deep learning is favoured in
recent years in view of its success in object detection and image classification. Several convolutional neural networks (CNNs) have been proposed to localize solar PVs from satellite imagery and estimate their sizes (Golovko et al. 2017; House et al. 2018; Liang et al. 2020; Malof et al. 2015). For example, Yu et al. (2018) utilized the transfer learning to train a CNN classifier for PV identification, then added an additional CNN branch directly connected to the intermediate layers for PV segmentation. Apart from the structure of deep networks, the quality of labelled samples largely determine the final accuracy
of obtained information (Ball et al. 2017; Reichstein et al. 2019). Researchers have spent a huge amount of time on building datasets generated from aerial or satellite imagery (Ji et al. 2019; Li et al. 2020; Xia et al. 2018). However, to date, there are no open-source datasets available for PVs, and no relevant studies evaluating the generalization ability of deep learning from aerial data to satellite data, and vice versa.



To meet the requirements of deep learning for labelled samples, we built a PV dataset from satellite and aerial imagery with three different spatial resolutions (i.e., 0.8m, 0.3m and 0.1m). We tested the effectiveness of our datasets in extracting multi-scale PVs using the coarse satellite samples (0.8m) for concentrated PVs, the medium aerial samples (0.3m) for distributed ground PVs, and the high-resolution unmanned aerial vehicle (UAV) samples (0.1m) for fine-grained rooftop PVs. In addition, we evaluated the feasibility of deep networks for cross applications between satellite and aerial samples. Our dataset will contribute to a variety of PV applications in the future.

## 2 Sampling area and data sources

All PV samples are collected in Jiangsu province, China, covering a total area of 107,200 square kilometres (Fig. 1a). Located in the lower reaches of the Yangtze River and Huaihe River, the province is very flat, averaging only 12.3m above sea level. The land terrain is mostly made up of low lands and flat plains, with hills and mountains in the southwest and north (Fig. 1b). With the continuous economic development and population growth, the energy demand in Jiangsu province increases rapidly. The government was committed to energy transition by improving energy efficiency and promoting the use of green energy. A number of policies were introduced to popularize solar PVs. Due to the shortage of land resources, most of installed PVs in Jiangsu province are distributed in areas where land competition is not fierce (e.g., sparse shrubs, low-density grasslands, reservoirs, ponds, saline alkali lands and rooftops), which makes it convenient to collect various PVs with different backgrounds.

The sizes of distributed PVs typically vary from a few panels to several hectares, depending on the area of available background land. It is difficult to identify all these PVs from a single data source; hence, we used satellite and aerial images with different spatial resolutions to collect PV samples at various scales. Gaofen-2 and Beijing-2 satellite images are used to prepare samples of large-scale PVs. Gaofen-2 is part of the CHEOS (China High Resolution Earth Observation System) family, and is capable of acquiring images with a ground sampling distance (GSD) of 0.81m in panchromatic and 3.24m in multispectral bands. Beijing-2 satellite constellation consists of three satellites, and can provide images with a GSD of 0.80m in panchromatic and 3.2 m in blue, green, red and near infrared bands. Aerial imagery with a GSD of 0.3m is used to collect samples of ground distributed PVs. The aerial photography was conducted by the Provincial Geomatics Centre of Jiangsu in 2018, covering the whole province. UAV images are used to collect rooftop PV samples. The UAV flight was carried out in Hai'an County (yellow box in Fig. 1b), where the development of rooftop PVs is relatively mature. Ground control point (GCP) data obtained by continuous operating reference stations were used for georeferencing. The final orthophotos have a GSD of 0.1m and location accuracy of approximately 0.02m. Fig. 1c-d illustrate the appearance of two rooftop PVs in different images. In Gaofen-2 image, the PVs take up only a dozen of pixels that are mixed with surrounding rooftops (Fig. 1c). It is difficult to distinguish the PVs from the backgrounds, let alone get their exact position and size. In contrast, PV detection becomes slightly easier in the aerial photograph (Fig. 1d), but obtaining accurate PV boundaries is still difficult. In the UAV image (Fig. 1e), we can clearly recognize the PVs, obtain their boundaries, and even count how many panels each

PV is composed of. This example illustrates the necessity of using multi-resolution images to build PV datasets that meet the needs of a variety of applications.

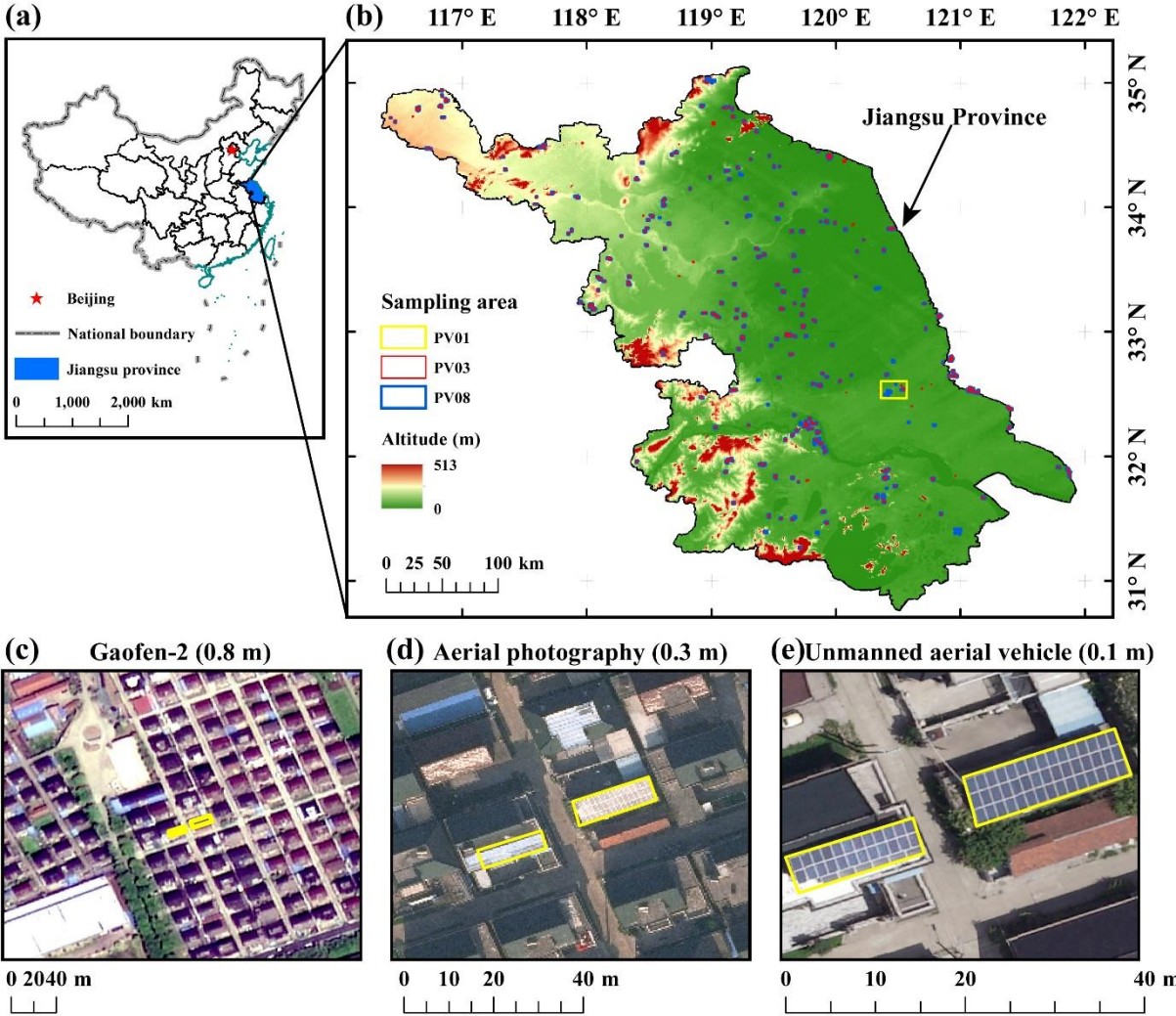

**Figure 1: Sampling area and data sources. (a) The location of Jiangsu province in China; (b) spatial distribution of all sampling areas; (c) Gaofen-2 satellite image with a spatial resolution of 0.8m; (d) image from aerial photography with a spatial resolution of 0.3m; and (e) image from unmanned aerial vehicle with a spatial resolution of 0.1m. The yellow boxes in sub-figure (c-e) represent the same rooftop PVs.**

## 3 Generation of PV samples

The schematic workflow to generate PV samples is shown in Fig. 2. The main procedures are described in the following:

1) *Data pre-processing*. To obtain high-quality PV samples, a series of pre-processing methods were applied to the original satellite and aerial images. We first checked the raw data and removed images with lots of clouds, noise and bright spots.



Geometric correction was undertaken to eliminate the spatial distortions in original images, and additional ortho-rectification was used for aerial images to minimise the perspective (tilt) and relief (terrain) effects. The adaptive Pan sharpening method (Song et al. 2016) was utilized to improve the spatial resolution of multi-spectral images by fusing the panchromatic band.

We also performed block adjustment on multi-temporal images to ensure that they have the same location accuracy. Finally, we use histogram equalization to adjust the hue component of the images.

2) *Category design*. Our PV dataset includes three groups of PV samples collected at different spatial resolutions (Table 1), namely PV08 from Gaofen-2 and Beijing-2 imagery, PV03 from aerial photography, and PV01 from UAV orthophotos. PV08 contains rooftop and ground PV samples. Ground samples in PV03 are divided into five categories according to their

background land use type: shrub land, grassland, cropland, saline-alkali, and water surface. Rooftop samples in PV01 are divided into three categories according to their background roof type: flat concrete, steel tile, and brick.

3) *Image annotation*. Due to the differences in the shape, size, and direction of various PVs, we used polygonal annotations, that is, drawing lines by placing points around the outer edges of each PV panel. The inner space surrounded by the points was then assigned a predefined code in Table 1 to indicate the category to which it belongs. The annotators worked in pairs

to ensure that each PV panel was annotated twice. After getting the initial annotations, a third annotator would merge the two annotations and check one by one to fix the potential errors. Finally, a supervisor was responsible for checking the quality of all annotations, including location and category. Figure 3 shows some examples of PV panels and their annotations.

4) *Sample making*. The shapefile of polygonal annotations was converted to a raster that has the same spatial resolution as satellite or aerial images. The raster and original red, green and blue (RGB) images were then seamlessly cropped into tiles

at a fixed size by referring to the sampling grids. Tiles containing a single category of PV were paired with corresponding image blocks to form a complete sample (refer to the example in Fig. 2). We prepared PV08 and PV03 samples at the size of 1024×1024 pixels, while PV01 samples at the size of 256×256 pixels. The numbers of each category are listed in Table 1.

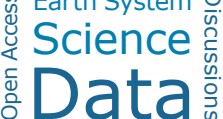

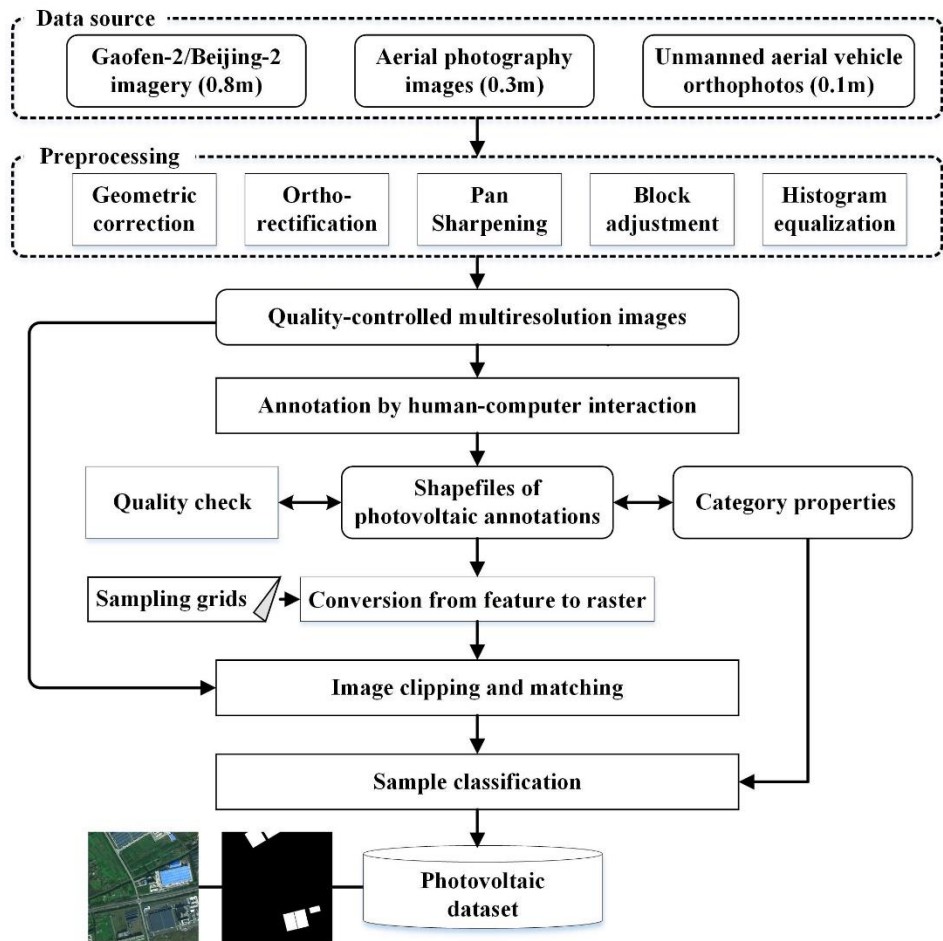

**Figure 2: Flowchart to generate PV samples.**

**Table 1: Classification system of our PV dataset.**

| Dataset | Category | Spatial Resolution | Code | Size | Num. |
|---|---|---|---|---|---|
| **PV08** | PV08_Rooftop | ~0.8m | 11 | 1,024×1,024 | 90 |
| | PV08_Ground | ~0.8m | 12 | 1,024×1,024 | 673 |
| **PV03** | PV03_Rooftop | ~0.3m | 111 | 1,024×1,024 | 236 |
| | PV03_Ground_Shrubwood | ~0.3m | 121 | 1,024×1,024 | 119 |
| | PV03_Ground_Grassland | ~0.3m | 122 | 1,024×1,024 | 117 |
| | PV03_Ground_Cropland | ~0.3m | 123 | 1,024×1,024 | 859 |
| | PV03_Ground_SalineAlkali | ~0.3m | 124 | 1,024×1,024 | 352 |
| | PV03_Ground_WaterSurface | ~0.3m | 125 | 1,024×1,024 | 625 |
| **PV01** | PV01_Rooftop_FlatConcrete | ~0.1m | 211 | 256×256 | 413 |
| | PV01_Rooftop_SteelTile | ~0.1m | 212 | 256×256 | 94 |
| | PV01_Rooftop_Brick | ~0.1m | 213 | 256×256 | 138 |

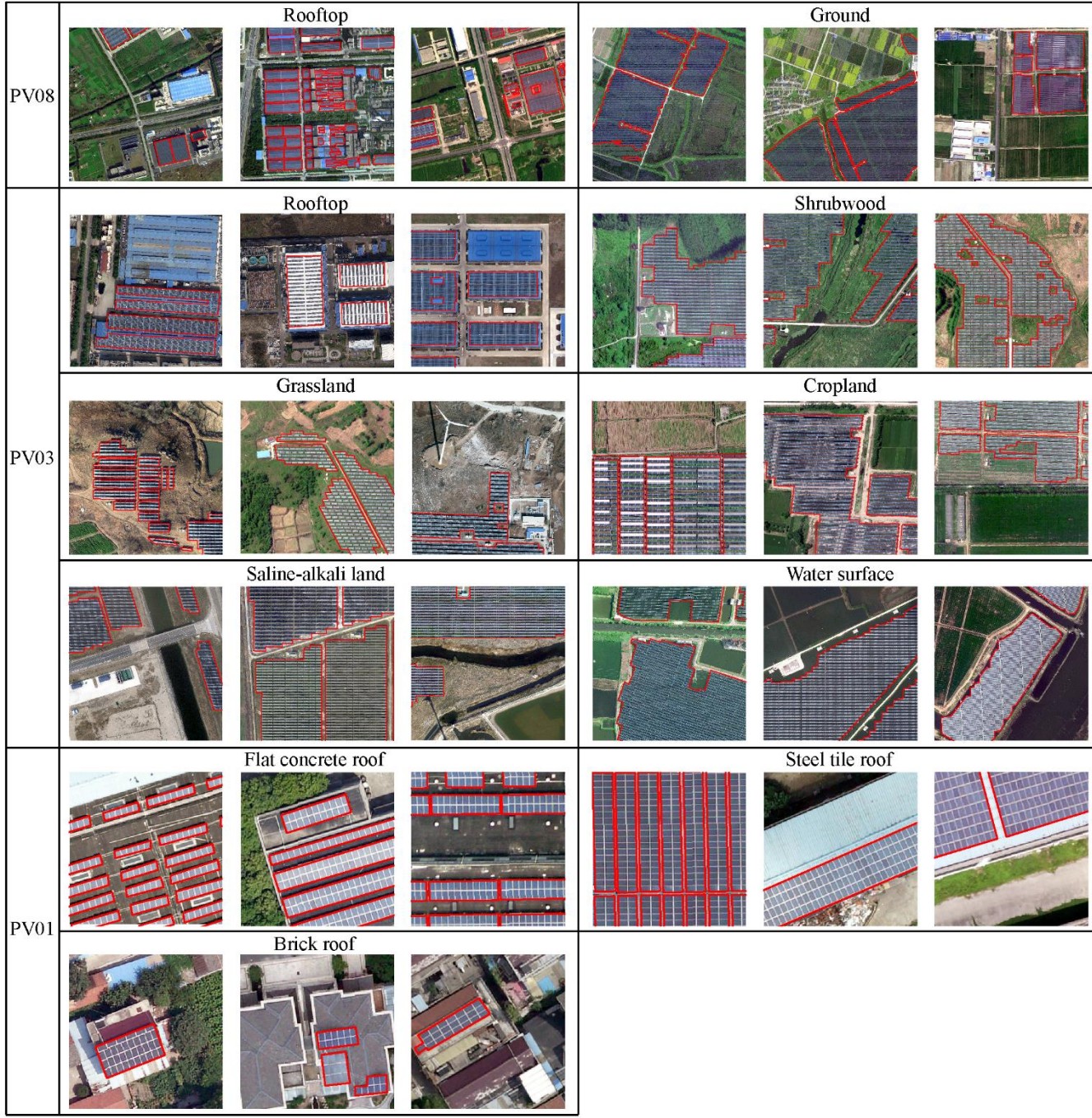

**Figure 3: Examples of PV panels and their annotations. Red boxes are the manually drawn boundaries of PV panels.**



## 4 Applications of the dataset

### 4.1 PV segmentation using deep networks

To examine the possibility of extracting multi-scale PVs from complex backgrounds based on our dataset, we carried out a group of segmentation experiments using deep learning. We compared the performance of three deep networks, including U-Net (Ronneberger et al. 2015), RefineNet (Lin et al. 2017) and DeepLab v3+ (Chen et al. 2018). The U-Net consists of a contracting path (encoder) to capture context and a symmetric expanding path (decoder) that enables precise localization. The feature map of the encoder is combined with the up-sampling feature map of the decoder through skip connection to

generate final segmentation map. The RefineNet is a multi-path refinement network, which exploits all information available along the down-sampling process to enable high-resolution prediction. The high-level semantic features are refined using low-level fine-grained features. In addition, a chained residual pooling is introduced into individual residual connections to capture background context. The DeepLab v3+ combines the advantages from spatial pyramid pooling module and encode-decoder structure. The former is capable of encoding multi-scale contextual information, while the latter can enhance the

ability to capture object boundaries.

The experiments were conducted on PV08, PV03 and PV01 dataset, respectively. For each category, all samples were separated into 80% training set (from which 20% samples were used for validation) and 20% testing set. The Adam optimizer was used for training and an early-stopping mechanism was adopted to prevent overfitting. The final segmentation results were evaluated using five indicators, including accuracy, precision, recall, F1 score, and intersection over union (IoU).

Accuracy refers to the ratio of PV and background correctly classified by the model to the sum of PV and background in the image. Precision is the ratio of PV correctly identified by the model to the total PV identified by the model, describing the reliability of PV segmentation results. The recall equals the ratio of PV correctly identified by the model to the actual total PV. F1 score ($\frac{2\times\text{precision}\times\text{recall}}{\text{precision}+\text{recall}}$) is a weighted average of precision and recall, providing a comprehensive evaluation of PV extraction result. IoU is the ratio of the intersection to the union between PV identified by the model and the actual PV. The

evaluation accuracy of PV segmentation results is summarized in Table 2. It is noted that different networks were compared under equal conditions, and additional techniques (e.g., data augmentation, class weight) were not taken into account.

**Table 2. Segmentation accuracy in terms of different evaluation indices.**

| Dataset | Model | Accuracy | Precision | Recall | F1 score | IoU |
|---------|-------|----------|-----------|--------|----------|-----|
| **PV08** | U-Net | 0.980 | 0.871 | 0.864 | **0.868** | 0.776 |
| | RefineNet | 0.979 | 0.848 | **0.884** | 0.866 | 0.773 |
| | DeepLab v3+ | **0.984** | **0.877** | 0.857 | 0.867 | **0.790** |
| **PV03** | U-Net | 0.973 | 0.897 | 0.935 | 0.916 | 0.858 |
| | RefineNet | 0.976 | 0.957 | **0.937** | **0.947** | 0.878 |
| | DeepLab v3+ | **0.983** | **0.959** | 0.931 | 0.945 | **0.908** |
| **PV01** | U-Net | 0.961 | 0.831 | **0.900** | 0.864 | 0.787 |
| | RefineNet | 0.981 | 0.909 | 0.897 | 0.903 | 0.859 |
| | DeepLab v3+ | **0.983** | **0.928** | 0.894 | **0.911** | **0.868** |

Overall, DeepLab v3+ achieved the highest accuracy across all three datasets, followed by RefineNet and U-Net. The disparity among different models was relatively small at coarse spatial resolution (approximately 2% in terms of IoU),
but the advantage of complex network became obvious as the spatial resolution increases (IoU difference reaches 5% for PV03 and 8% for PV01). The reasonable explanation is that in coarse satellite images the blurred boundaries between PV and background prevent the complex networks from acquiring more useful information. Figs. 4–6 show some examples, which helps in understanding the effects of network structure and image resolution on the final segmentation results. With respect to the results of DeepLab v3+, some parts of PV were lost (e.g., Figs. 4d, 5d and 6c) and the gaps between adjacent
PVs were wider than the actual (e.g., Figs. 4b, 5d and 6b). In contrast, RefineNet and U-Net misclassified portions with similar characteristics as PV (e.g., Figs. 4a, 4b, 4d, 5a, 5c, 5f, 6b and 6c). The phenomena suggest that DeepLab v3+ tends to ensure the extracted PVs are reliable, while RefineNet and U-Net try to identify all PVs as many as possible. This explains why the precision of DeepLab v3+ was superior to those of RefineNet and U-Net, but the recall was the opposite (Table 2).

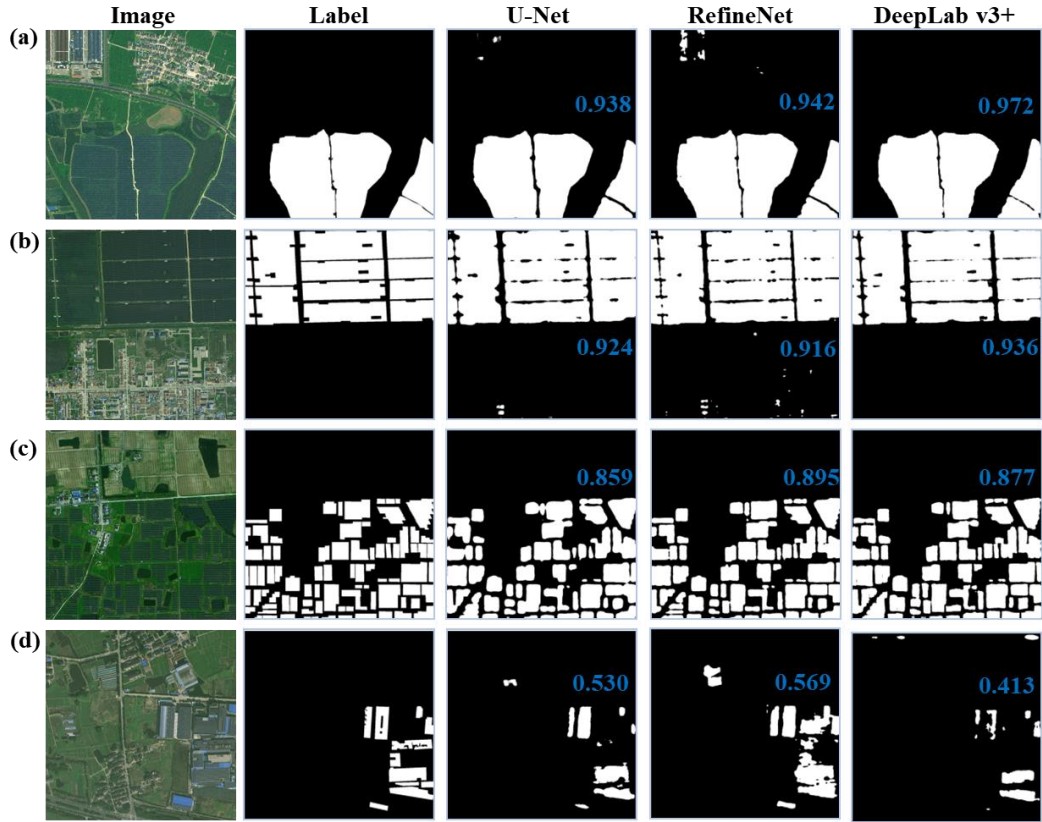

**Figure 4: Segmentation results of PVs in PV08 dataset. We show examples of concentrated ground PVs (a, b), distributed ground PV (c), and distributed rooftop PV (d). The IoU of each segmentation result is marked in blue within the image.**

**Figure 5: Segmentation results of PVs in PV03 dataset. Examples correspond to PV on shrub land (a), grassland (b), cropland (c), saline-alkali (d), water surface (e), and rooftop (f), respectively. The IoU of each segmentation result is marked in blue within the image.**



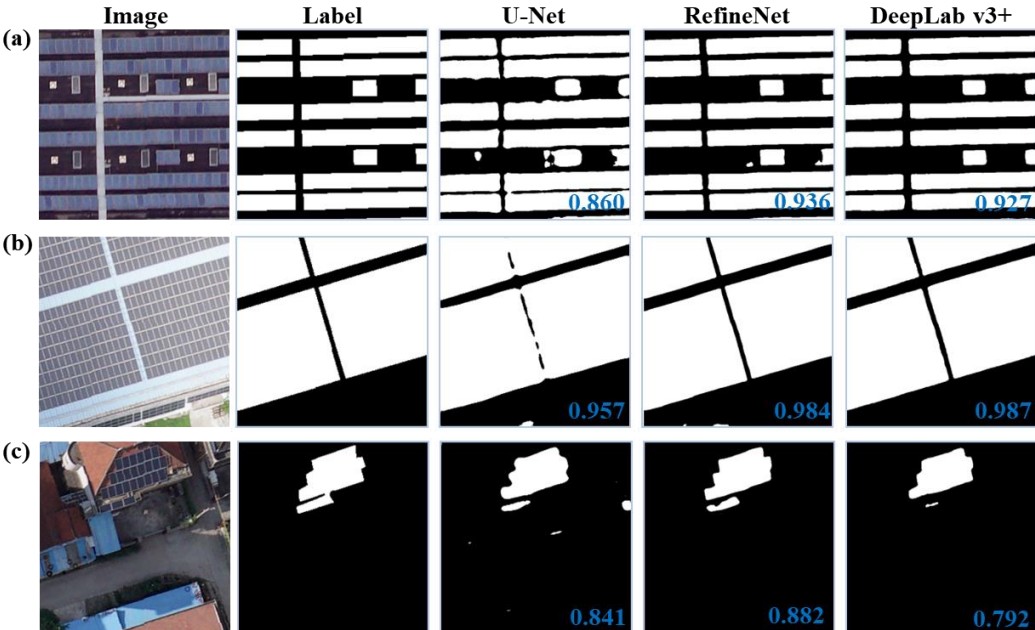

**Figure 6: Segmentation results of PVs in PV01 dataset. Examples corresponds to PV on flat concrete (a), steel tile (b) and brick (c) roofs, respectively. The IoU of each segmentation result is marked in blue within the image.**

Utility-scale PVs account for approximately 88% of the samples in PV08. The unbalance of training samples led to the difference in segmentation accuracy (higher for utility-scale PVs while lower for distributed PVs, Fig. 4). Except that, the spatial resolution was responsible for the poor performance on distributed PVs (Fig. 4c–d) that were mixed with background in the 0.8m satellite images. We may conclude that PV08 samples are only suitable for large-scale PV extraction, and higher resolution is required for distributed PVs. Intuitively, the texture of distributed PV becomes clear in the 0.3m aerial images, and the contrast to background is significant, making it easy to distinguish PV from various backgrounds. The average IoU of DeepLab v3+ reached 0.900, 0.884, 0.920, 0.903, 0.911, and 0.926 for PVs on shrub land, grassland, cropland, saline-alkali, water surface, and rooftop, respectively, which revealed that the segmentation accuracy was slightly affected by the background land types. PVs on flat concrete and steel tile roofs occupy the entire roof of large buildings, such as factories, shopping malls, business centres and urban residential buildings, thus seem "large-scale" in the UAV images with a spatial resolution of 0.1 m. On average, DeepLab v3 achieved an IoU of 0.873 for flat concrete PVs and 0.927 for steel tile PVs. In contrast, PVs on brick roofs of rural residential building and urban villa usually consist of several panels because the limited area available for PV installations. These "small-scale" PVs may share the same feature with surrounding roofs or shadows, thus the segmentation accuracy was reduced to 0.850 in terms of IoU. Based on the above analysis, we recommend PV08 for extracting concentrated PVs, PV03 for ground distributed PVs, and PV01 for rooftop distributed PVs.





## 4.2 Cross application at different resolutions

The generalization capability of deep learning is critical to automatic information extraction. This section investigates the feasibility of cross application between PV samples with different spatial resolutions, including between PV08_Ground and PV03_Ground, and between PV03_Rooftop and PV01_Rooftop. We compared the segmentation results of DeepLab v3+ from direct training, cross application and fine-tuning. Taking the experiment between PV08_Ground and PV03_Ground as an example, direct training means that DeepLab v3+ trained on PV08 (PV03) samples was applied to PV08 (PV03) samples; cross application means that the model was trained on PV03 (PV08) samples but applied to PV08 (PV03) samples; and fine-tuning means that the model was first pre-trained on PV03 (PV08) samples, then fine-tuned (fine-tuning process lasted 10 epochs) using PV08 (PV03) samples, and finally applied to PV08 (PV03) samples.

According to Table 3, the segmentation accuracy of cross application was terrible with extremely low recall and IoU. After fine tuning, the accuracy increased rapidly to a level comparable to direct training. Some examples are given in Figs. 7–8, where the feature maps indicating the probability that each pixel belongs to PV are illustrated for cross application and fine-tuning experiments. It can be seen that during cross application, the model captured the main feature of PV, but the difference between PV and background was not significant. Through fine-tuning, the differences were enhanced; hence, PV could be easily segmented. Our experiments demonstrate that there are inherent defects in the cross application at different resolutions, but these defects can be compensated by fine-tuning on target dataset. The fine-tuning approach avoids the time consumption of direct training and the huge investment of building compelete datasets with various resolutions.

**Table 3. Segmentation accuracy of DeepLab v3+ trained by different strategies**

| Dataset | Model | Accuracy | Precision | Recall | F1 score | IoU |
|---|---|---|---|---|---|---|
| PV08_Ground | Direct training | **0.984** | **0.907** | 0.908 | **0.908** | **0.845** |
| | Cross application | 0.935 | 0.856 | 0.517 | 0.645 | 0.492 |
| | Fine tuning | 0.978 | 0.867 | **0.922** | 0.894 | 0.823 |
| PV03_Ground | Direct training | **0.981** | **0.960** | **0.903** | **0.931** | **0.877** |
| | Cross application | 0.752 | 0.726 | 0.185 | 0.295 | 0.177 |
| | Fine tuning | 0.975 | 0.943 | 0.897 | 0.919 | 0.865 |
| PV03_Rooftop | Direct training | 0.977 | 0.824 | **0.823** | 0.824 | 0.707 |
| | Cross application | 0.894 | 0.414 | 0.048 | 0.086 | 0.045 |
| | Fine tuning | **0.981** | **0.891** | 0.811 | **0.849** | **0.747** |
| PV01_Rooftop | Direct training | **0.983** | **0.928** | **0.894** | **0.911** | **0.868** |
| | Cross application | 0.846 | 0.672 | 0.403 | 0.504 | 0.368 |
| | Fine tuning | 0.965 | 0.918 | 0.809 | 0.860 | 0.784 |

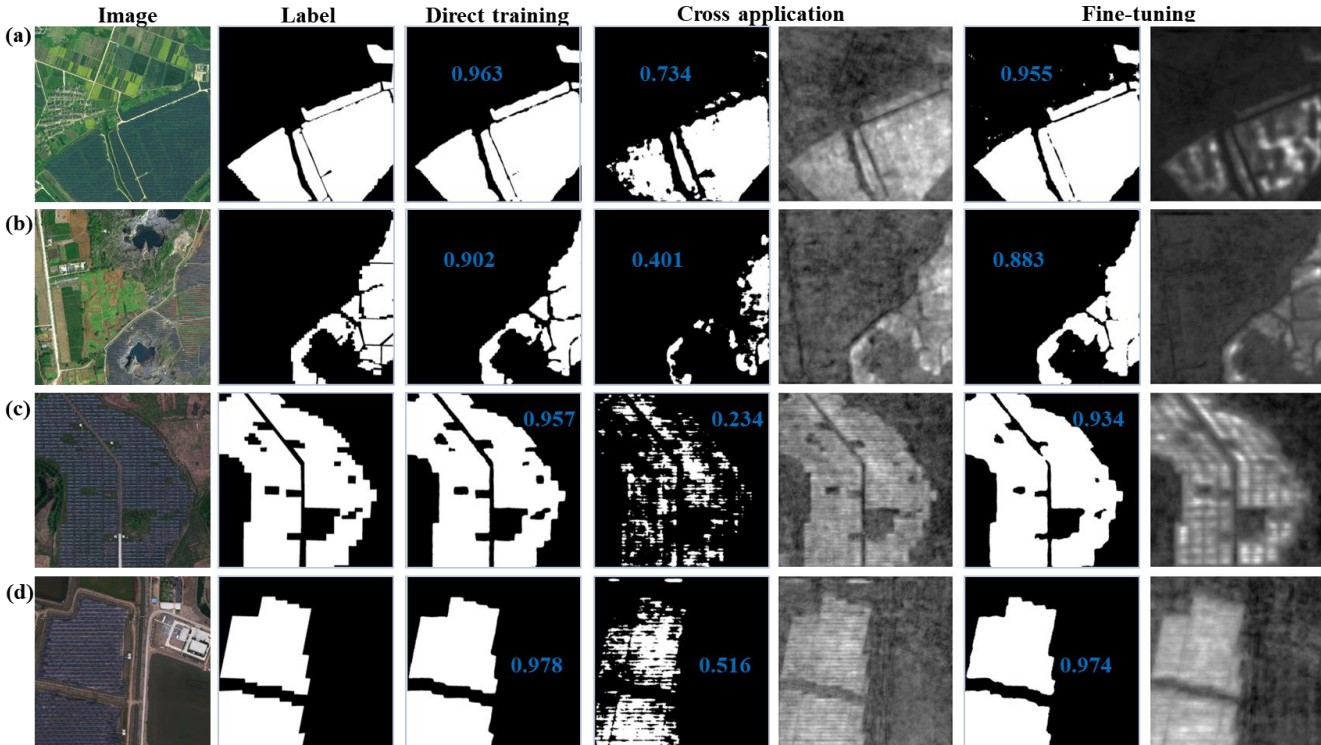

Figure 7: Cross application of ground PV samples. Segmentation results of DeepLab v3+ from direct training, cross application and fine-tuning are shown for PVs in PV08 (a, b) and PV03 (c, d) dataset. Feature map for cross application and fine-tuning is displayed on the right of corresponding segmentation result. IoU of each segmentation result is marked in blue within the image.

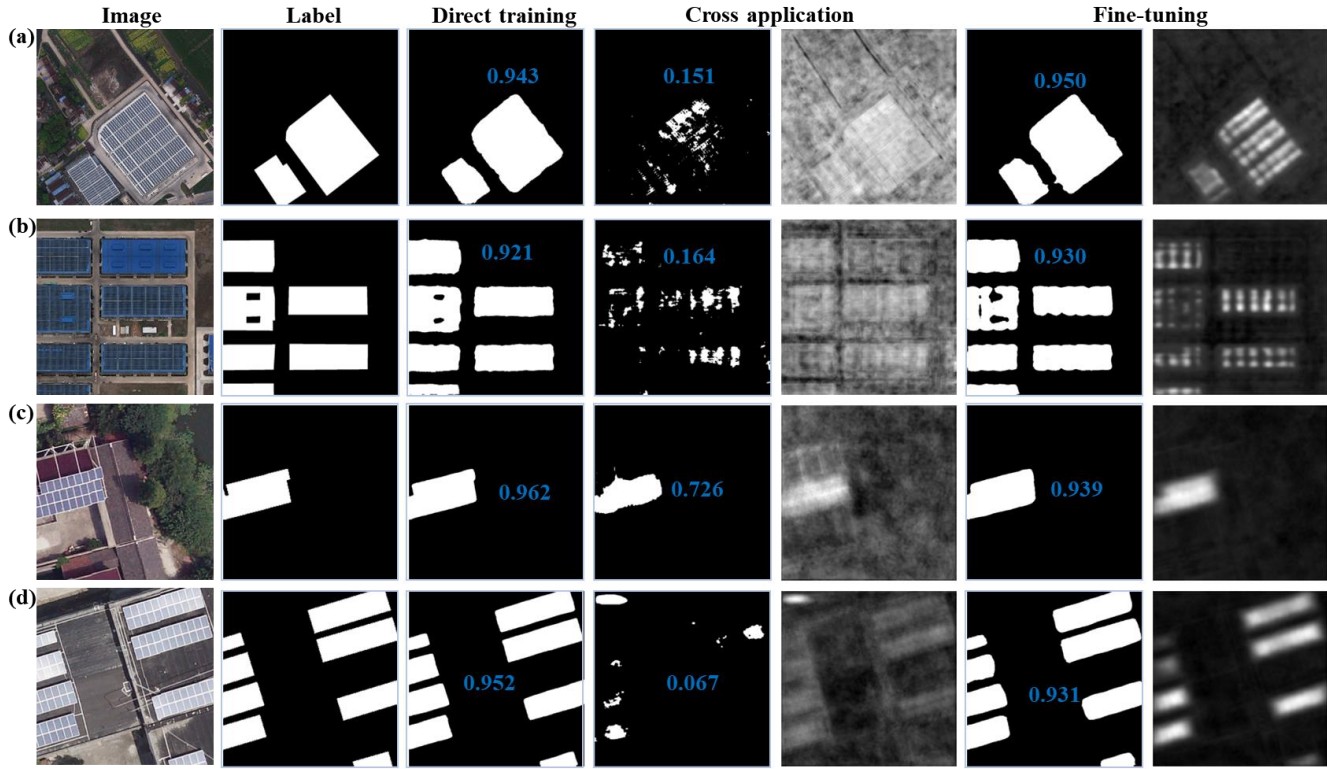

**Figure 8: Cross application of rooftop PV samples. Segmentation results of DeepLab v3+ from direct training, cross application and fine-tuning are shown for PVs in PV03 (a, b) and PV01 (c, d) dataset. Feature map for cross application and fine-tuning is displayed on the right of corresponding segmentation result. IoU of each segmentation result is marked in blue within the image.**

## 5 Data availability

The PV dataset is freely available from the Zenodo website at https://doi.org/10.5281/zenodo.5171712 (Jiang et al. 2021). There are three compressed folders, namely PV08.zip, PV03.zip and PV01.zip, for PV samples collected at the spatial resolution of 0.8m, 0.3m and 0.1m, respectively. The original images are named as "PV0*_XXXXXX_YYYYYYY.bmp" and the corresponding labels are named as "PV0*_XXXXXX_YYYYYYY_label.bmp" (* can be the number 8, 3 or 1). The central location (latitude, longitude) of each image equals (XX.XXXX, YY.YYYYY). For each label, "0" indicates the background while the target PV is recorded as the code listed in Table 1.

## 6 Conclusions

This study built a multi-resolution dataset for PV panel segmentation, including PV08 from Gaofen-2 and Beijing-2 satellite image with spatial resolution of 0.8m, PV03 from aerial images with spatial resolution of 0.3m, and PV01 from UAV images with spatial resolution of 0.1m. Samples cover a variety of PVs installed on different lands (i.e., shrub land, grassland,



cropland, saline-alkali, and water surface) and various rooftops (i.e., flat concrete, steel tile, and brick roofs), ranging in size from dozens of panels to several hectares. To the best of our knowledge, this is the first open PV dataset with multiple spatial resolutions.

Based on the dataset, we investigated the performance of different deep networks on PV segmentation and evaluated the feasibility of cross application between different resolutions. It is recommended to use PV08 for concentrated PV, PV03 for distributed ground PV, and PV01 for distributed rooftop PV so as to achieve the best segmentation results with an IoU of 0.845, 0.871 and 0.868, respectively. It also proved that direct cross applications do not work well and fine-tuning of pre-trained network using the target samples is essential. Besides, this dataset can contribute to various research and applications related to PV.

**Author contributions.** Hou Jiang: Methodology, Formal analysis, Writing – original draft. Ling Yao: Conceptualization, Writing – review & editing, Funding acquisition. Ning Lu: Visualization, Writing – review & editing. Jun Qin: Software, Investigation. Tang Liu: Validation, Data Curation. Yujun Liu: Resources, Data Curation. Chenghu Zhou: Supervision, Project administration.

**Competing interests.** The authors declare that they have no conflict of interest.

**Acknowledgements.** This work was supported by the National Natural Science Foundation of China (No.41771380), and the Key Special Project for Introduced Talents Team of Southern Marine Science and Engineering Guangdong Laboratory (No. GML2019ZD0301). We are grateful to the Provincial Geomatics Center of Jiangsu for their assistance in processing satellite and aerial images, the GitHub user Attila94 for sharing RefineNet code (https://github.com/Attila94/refinenet-keras), and the GitHub user sunlinlin-aragon for sharing DeepLab v3+ code (https://github.com/sunlinlin-aragon/DeepLabV3_Plus-Tensorflow2.0).

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
