# Peer review of "Multi-resolution dataset for photovoltaic panel segmentation from satellite and aerial imagery"

_Earth System Science Data, 2021_

## Author Response (AR1)

Dear Editor and reviewers:

Thank you for your letter and comments concerning our manuscript. Those comments are all valuable and very helpful for improving our paper, as well as the important guiding significance to further research. We have studied comments carefully and made correction which we hope meet with your approval. Our draft is revised in Revised Mode of Microsoft Word to ensure the revised portion obvious. Our responses are shown in "Blue" color and the changes in the manuscript are shown in "Red" color. In the following, the point-to-point response to the Referees, the draft in Revised Mode, and the clean version of our manuscript, are present in order.

**Response to Referee #1**

**9 Summary and comments:**

This paper presents a dataset as a benchmark for developing and evaluating methods to extract PV panels from satellite 11 images. Such a dataset is scarce but timely for the relevant community and will help researchers reduce the time and cost of collecting high-quality samples and will enrich other work related to solar PV energy. Overall, this manuscript is technically sound and well-crafted.

I would like to suggest a revision before it can be considered for publication in the Earth System Science Data.

Response:

We thank Referee #1 for the encouraging comments. All comments and suggestions have been considered carefully and welladdressed.

**19 Comments:**

1. The dataset was collected in Jiangsu Province. Considering that the changes in geographic context may affect the model's
 performance, how can you guarantee that the model trained with this dataset can be generalized to other regions?

**22 Response:**

We compared the samples from Gaofen-2 and Beijing-2 images, and found that PV panels exhibit similar characteristic in 23 24 high-resolution imagery and that the main difference comes from the background. The changes in geographic context will 25 inevitably affect the performance of deep networks. To mitigate such effects, we tried to collect PV samples covering as 26 many backgrounds as possible to enhance the generalization ability of deep networks trained by our dataset. The installed 27 PVs in Jiangsu province are distributed on various land covers, such as, sparse shrubs, low-density grasslands, reservoirs, ponds, saline alkali lands and rooftops, making our dataset representative for most cases. Besides, some skills in deep 28 29 learning community can be adopted to guarantee the generalization ability in other regions, such as transferring learning, 30 cross-domain feature representation. In completely different areas where direct application may fail, using a small number of 31 samples in the target area to fine-tune the pre-trained model can also achieve satisfactory results. Therefore, although the 32 samples were collected in Jiangsu Province, there is no need to worry too much. Furthermore, our work will continue, and 33 we plan to collect more PV samples in China and around the world through automatic deep learning algorithm and manual 34 discrimination. We have added one paragraph at the end of Section 3 to make this point clear: One concern of our data set is 35 the representativeness of the samples because the changes in geographic context will inevitably affect the performance of 36 deep learning models. We compared the samples from Gaofen-2 and Beijing-2 images, and found that PV panels exhibit 37 similar characteristic in high-resolution imagery and that the main difference comes from the background. Therefore, we 38 collected samples covering as many backgrounds as possible to ensure the representativeness. Besides, some skills (e.g., 39 transferring learning, cross-domain feature representation) in the deep learning community can be adopted to enhance the 40 generalization ability of deep networks trained by our dataset, which is beyond the discussion of this study. In the following, 41 we introduce some applications of deep learning to illustrate the quality and value of our dataset.

**43 Comments:**

2. The dataset is composed of Gaofen-2 and Beijing-2 imagery, which include a near-infrared band in addition to RGB bands.
Does this mean that the model trained with this dataset can only be applied to the satellite imagery with a specific combination of spectrums? Are these two imagery representative of the current common satellite imagery?

**47 Response:**

Actually, all samples in our dataset are composed of red, green and blue (RGB) bands of used satellite and aerial images. Our experiments (Section 4.1) also demonstrate that RGB bands are enough for deep networks to distinguish PV panels from various backgrounds. Therefore, the near-infrared band of Gaofen-2 and Beijing-2 images is not contained in our samples. We have made it clear in Section 3 as "The shapefile of polygonal annotations was converted to a raster that has the same spatial resolution as satellite or aerial images. The raster and original red, green and blue (RGB) images were then seamlessly cropped into tiles at a fixed size by referring to the sampling grids."

The comparison of PV samples from Gaofen-2 and Beijing-2 imagery shows that different PV panels exhibit similar characteristic after histogram equalization. Thus, we believe that the two satellite imagery are representative for common applications. We have made it clear in Section 3 as "One concern of our data set is the representativeness of the samples because the changes in geographic context will inevitably affect the performance of deep learning models. We compared the samples from Gaofen-2 and Beijing-2 images, and found that PV panels exhibit similar characteristic in high-resolution imagery and that the main difference comes from the background. Therefore, we collected samples covering as many backgrounds as possible to ensure the representativeness." 62 Summary and comments: 63 This research focused on validating deep learning as a tool to automatically extract photovoltaic panels from satellite and 64 aerial imagery with various spatial resolutions. Using RGB bands from images, the study received a high accuracy in the 65 classification and segmentation of PV panels, and the transferability of the models trained with different resolution samples was also discussed. Generally, the proposed approach was appropriate for the aim of this study. I have some concerns and 66 67 suggestions. **Response:** 68 Thank you for your encouraging comments. We have studied your suggestions and made correction which we hope meet 69 with your approval. The response is present in the following. 70

**72 Comments:**

1 – According to the classification system introduced in Table 1, the PV dataset introduced in this study may including 74 different categories of PVs. However, in the result part, all the categories were classified as a whole. The authors may want 75 to show the influence of different resolutions on classifying different categories of PVs, but the "classification system" used 76 here may confuse. My suggestion is to change the term "classification system" in the title of Table 1.

**77 Response:**

Thank you for your suggestion. In order to support more applications and give full play to the data value, we divide all samples into different sub-categories according to their background land use or roof type. In the result part, we focus on validating the impact of different resolutions on PV classifications; hence, all sub-categories are classified as a whole. To avoid confusion, the term "classification system" is replaced by "organizational structure" in the revised manuscript, that is, the revised title of Table 1 is "Organizational structure of our PV dataset."

**84 **Comments:**

– The names of segmentation networks in the manuscript should be checked. For example, "DeepLab v3+" was taken as
"DeepLab v3" in Line. 189, Page. 11.

**87 Response:**

Thank you very much. We have carefully checked and corrected the names of the three segmentation networks in the 89 manuscript. "DeepLab v3" should be "DeepLab v3+" and the revised sentence is "On average, DeepLab v3+ achieved an 90 IoU of 0.873 for flat concrete PVs and 0.927 for steel tile PVs."

**61**

**Response to Referee #2**

**93 Comments:**

3 – This study gave suggestions on the selection of image resolutions for the classification of different PVs. The differences 95 in the classification results of images with different resolutions may be related to the size of the target features and input 96 samples, because the semantic segmentation networks are generally sensitive to the size, shape, and receptive fields. It may 97 be interesting to give a quantified result on selecting image resolutions and input sample sizes for target features with 98 different sizes in the future study.

**99 Response:**

Thank you for your suggestions. In this study, we recommend to use PV08 for concentrated PV, PV03 for distributed ground 101 PV, and PV01 for distributed rooftop PV so as to achieve the best segmentation results. However, the quantitative result on 102 selecting image resolutions and input sample sizes for PVs with different sizes might require samples at more resolutions. 103 Therefore, this interesting suggestion is only discussed in Section Conclusion of the revised manuscript. The related part is 104 "Besides, this dataset may contribute to a diversity of other research and applications related to PV. For example, the 105 segmentation networks are generally sensitive to the observational size and shape in the receptive field; hence, it is valuable 106 to quantitatively explore the general guidelines on selecting image resolutions and input sample sizes for PVs with different 107 sizes."

**109 Comments:**

4 – Authors used images with different resolutions to extract PVs in the study area. However, it may be interesting to see the 111 classification results when combining multiscale features from images with different resolutions. The fusion of multiscale 112 features can be further discussed in the future.

**Response:**

Thank you for your suggestions. This paper presents two simplest applications based on our PV dataset. Now that multiresolution dataset has been established, more complex cases are feasible. We hope that in the future our dataset will support more valuable and interesting research, for example, investigating whether a network can be established to fuse multiscale features from images with different resolutions to achieve synchronous identification or segmentation of multi-scale PVs. We have added this point into the Conclusion in the manuscript as "Whether a network can be established to combine images with different resolutions to achieve synchronous identification of multi-scale PVs is also of great interest."

**Response to Referee #3** 122 123 Summary and comments: 124 This paper introduces a multi-resolution PV dataset that composes of various samples collected from satellite and aerial 125 images. Such a dataset is of great interest to users because it helps to develop deep learning algorithms for automatic PV 126 information extraction. The paper is well organized and the dataset is described in a clear fashion. Therefore, I would like to 127 suggest for publication in the Earth System Science Data after a minor revision. 128 **Response:** 129 Thank you for your encouraging comments. 130 131 **Comments:** 132 1) For the segmentation experiments (Section 4.1), the samples are divided into 80% training set and 20% testing set. Is the 133 division performed randomly for PV08/03/01 or separately for each subcategory? The authors should make it clear because 134 the two approaches lead to different model performance and segmentation results. 135 **Response:** 136 Thank you for your advice. Actually, the division is performed for each subcategory of PV08/03/01. For example, the 137 training (testing) set for the experiment on PV08 contains 80% (20%) samples from PV08 Rooftop and 80% (20%) samples from PV08 Ground. We have made it clear in the revised manuscript as "The experiments were conducted on PV08, PV03 138 139 and PV01 dataset, respectively. For each sub-category (e.g., PV08 Rooftop, PV08 Ground), all samples were separated into 140 80% training set (from which 20% samples were used for validation) and 20% testing set." 141 142 **Comments:** 143 2) For the cross application (Section 4.2), I wonder whether authors divide training samples and testing samples in the same 144 way as Section 4.1? The authors state that "fine-tuning means that the model was first pre-trained on PV03 (PV08) samples, 145 then fine-tuned (fine-tuning process lasted 10 epochs) using PV08 (PV03) samples, and finally applied to PV08 (PV03) 146 samples." Are the samples used for fine tuning the same as those used for direct training? If yes, how will the model perform if only using a small portion samples for fine-tuning.

**Response:**

- 149 For the cross application, we divide training set and testing set in the same way as Section 4.1. The samples used for fine
- 150 tuning are not the same as those used for direct training. Only a small portion (20%) samples from the training set of the
- 151 target PV dataset are selected for fine-tuning. We have made it clear in the revised manuscript as "The training set account
- 152 for 80% of the whole dataset and the testing set is the remaining 20%, but only 20% samples from the training set of the
- 153 target PV dataset are randomly selected for fine-tuning."